# Prevalence of visual impairment and its causes in adults aged 50 years and older: Estimates from the National Eye Surveys in Malaysia

Mohamad Aziz Salowi[1,2,3]☯, Nyi Nyi Naing📷[2]*, Norasyikin Mustafa[4]☯, Wan Radziah Wan Nawang[3]☯, Siti Nurhuda Sharudin[3]☯, Nor Fariza Ngah[5]

1 Clinical Research Centre (CRC), National Institute of Health (NIH), Shah Alam, Malaysia, 2 Faculty of Medicine, Universiti Sultan Zainal Abidin, Kuala Terengganu, Terengganu, Malaysia, 3 Department of Ophthalmology, Selayang Hospital, Ministry of Health, Putrajaya, Malaysia, 4 Faculty of Medicine, Universiti Teknologi MARA, Shah Alam, Selangor, Malaysia, 5 Deputy Director General Office (Research and Support), Ministry of Health, Putrajaya, Malaysia

☯ These authors contributed equally to this work.
* syedhatim@unisza.edu.my

**Data Availability Statement:** All survey data are available from the database URL https://www.raab.world/survey-data.

## Abstract

### Background

Population surveys are necessary to measure a community's eye care needs. We conducted simultaneous surveys in two regions in Malaysia in 2023 to estimate the prevalence of blindness and/or visual impairment (VI), identify its main causes, and compare the results with the survey in 2014.

### Methods

The surveys were simultaneously done in Eastern and Sarawak administrative regions using the Rapid Assessment of Avoidable Blindness (RAAB) technique. It involved a multi-stage cluster sampling method, each cluster comprising 50 residents aged 50 years and older. The prevalence of blindness and/or visual impairment (blindness, severe, moderate, and early) and its primary cause were determined through a visual acuity test and eye examination with a hand-held ophthalmoscope. Results were compared with the previous survey in 2014.

### Results

A total of 10,184 subjects were enumerated, and 9,709 were examined (94.5% and 96.2% responses for Eastern and Sarawak, respectively). The prevalence of blindness and severe VI appeared lower than the previous survey. For blindness: Eastern 1.4%, 95%CI (0.9, 1.9) to 0.8%, 95%CI (0.5, 1.1) and Sarawak: 1.6% 95%CI (1.0, 2.1) to 0.6%, 95%CI (0.3, 0.9). For severe VI: Eastern 1.2%, 95%CI (0.8, 1.7) to 0.9%, 95%CI (0.6, 1.1) and Sarawak 1.1% 95%CI (0.6, 1.6) to 0.9%, 95% CI(0.6, 1.2). The main cause of blindness was untreated

**Funding:** Mohamad Aziz bin Salowi Grant number:91000984 Ministry of Health Grant Sponsor does not play any role in the study design, data collection and analysis, decision to publish, or preparation of the manuscript.

**Competing interests:** The authors have declared that no competing interests exist.

cataracts: 77.3% (Eastern) and 75.0% (Sarawak). Diabetic retinopathy was the 2nd main cause of blindness for Eastern at 9.1%, but it only caused early to severe VI in Sarawak.

## Conclusion

The prevalence of blindness and severe VI were lower than in the previous survey. It could have been attributed to a community cataract program implemented soon after the survey in 2014. However, more efforts are needed to address the high percentage of avoidable blindness within both regions.

## Introduction

Malaysia is one of the 37 member states of the World Health Organisation (WHO) Western Pacific Region [1]. As part of the global and regional eye health agendas, Malaysia has been actively engaged with other countries within the Western Pacific Region in planning, implementing, and monitoring community programs related to the Prevention of Blindness and Low Vision (PBL) initiatives. The activities' coverage expanded and strengthened after the country endorsed and signed the World Health Assembly (WHA) 66.4 resolution in May 2013 [2,3]. By signing the resolution, Malaysia is committed to being part of the Global Action Plan, the core activity and aim of providing Universal Health Care (UHC) to the population [3].

In 2010, over 90 million people in the Western Pacific Region were visually impaired, including more than 10 million blind. Blindness or visual impairment can be avoided in 80% of cases with appropriate treatment or early prevention [4]. In Malaysia, the estimated prevalence of blindness in the country, according to the findings of the Six Simultaneous National Eye Survey, NES II (2014) was 1.2%. Untreated cataracts were the leading cause, followed by diabetic retinopathy and glaucoma. When estimated by individual PBL administrative regions (Fig 1), there were discrepancies between the regions: Northern (1.5%), Eastern (1.4%), Central (0.5%), Southern (0.9%), Sabah (1.9%) and Sarawak (1.6%) with untreated cataract remained the main cause of blindness across the regions [5].

One of the national action plans planned and implemented after the survey was introducing a community eye care program, which could potentially reach the underserved population, especially in remote areas. Eastern Region and Sarawak were selected as the regions where this program would be piloted because they had the necessary eye care ownership and network among the hospitals, the health state departments, the NGOs and the community leaders, which would ensure the sustainability of a community program. Each region had a Public Health Ophthalmologist who could monitor and champion the program.

This study aimed to determine the difference or improvement in the prevalence of blindness and/or visual impairment between the two administrative regions from the previous survey.

## Material and methods

A follow-up survey was required to compare the prevalence of blindness and/or visual impairment before and after the community eye care program's implementation. These cross-sectional, population-based surveys, which followed the World Health Organization (WHO)-recommended RAAB protocol, were conducted simultaneously from 27 July 2023 to 7 October

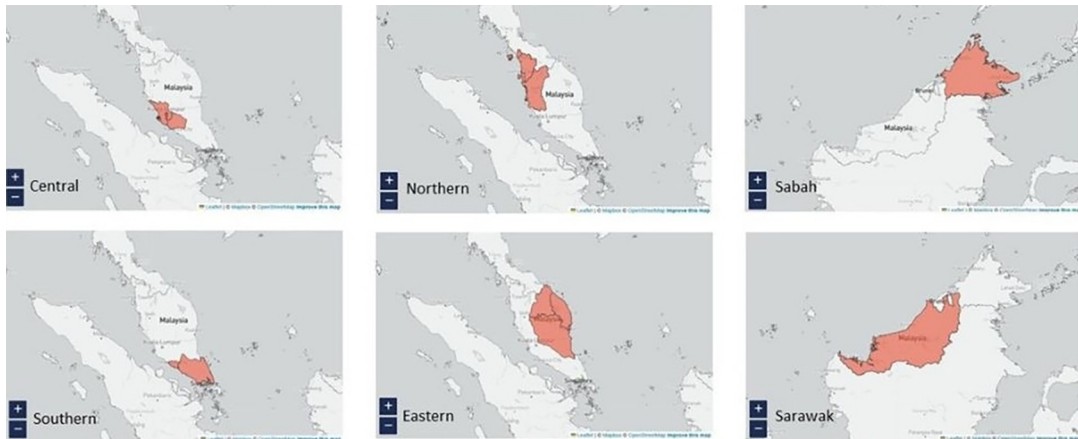

Source and Copyright: OpenStreetMap: https://www.openstreetmap.org. and RAAB world: https://www.raab.world/

| Region | Total Population (million) | Population 50 years and older (%) |
|---|---|---|
| **Eastern** | **4.6** | **13.0** |
| **Sarawak** | **2.5** | **8.6** |
| Central | 10.3 | 31.2 |
| Northern | 6.7 | 23.6 |
| Southern | 5.0 | 15.4 |
| Sabah | 3.5 | 8.1 |

Source and Copyright: Department of Statistic Malaysia 2020

**Fig 1. Survey administrative regions, total population, and percentage of individuals 50 years and older in Malaysia.** (This study surveyed Eastern and Sarawak only). Reprinted from [Rapid Assessment of Avoidable Blindness: Survey Data. Available from: https://www.raab.world/survey-dataRAAB world] under a CC BY license, with the permission from OpenStreetMap Foundation (OSMF), original copyright year 2024.

2023. The surveys also collected data on cataract surgical coverage (CSC) and cataract surgical outcomes, the results of which will be discussed in another manuscript.

Each region had six (6) data collector teams comprising three persons: two doctors and one allied health staff member trained in ophthalmology. Each team was responsible for surveying 16–17 randomly selected clusters, examining 50 residents aged 50 years and older. Population sampling was per RAAB methodology, a widely used, WHO-recommended method for population-based surveys of the prevalence of VI and its causes. The RAAB survey protocol and methodology have been described elsewhere [6].

### Sampling frame

Department of Statistics, Malaysia (DOSM) conducts nationwide data collection for the National Population and Housing Census once every ten years. An Enumeration Block (EB), the smallest population unit, with 80–120 living quarters each, is demarcated based on the latest findings and the population distribution, followed by the development of a corresponding geographical map, indicating the exact location and boundaries of each EB. The EBs are gazetted for field work operations, for example, Morbidity, Nutrition, Household Expenditure, and Labor Force Survey [7]. The complete list of all EBs from the 2020 national census was used to select clusters for the RAAB. A total of 105 EBs were randomly chosen for Eastern and

98 EBs for Sarawak regardless of strata, using the Probability Proportionate to Size (PPS) technique. Selected EB codes and the corresponding maps were then used to identify the location of the EBs during fieldwork data collection.

## Training

Training for survey teams was conducted separately in each region before the fieldwork (one training session in Sarawak and one training session in the Eastern Region). Those training sessions by a certified RAAB trainer were to ensure data quality and strict adherence to study protocol. Survey team members were required to attend four training days as preparation, including RAAB lectures, inter-observer variation assessment and a pilot survey in a test cluster in one of the nearby EB's during fieldwork. Day 1 and Day 2 of training focused on the RAAB7 data entry and management using tablets and also ocular examination exercises, including fundus examinations.

The third day of the training was fully dedicated to the inter-observer variation (IOV) assessment, exercise and retraining. Six survey teams were assessed in their agreements, asking 15 questions to subjects comprising patients and staff from the Ophthalmology Outpatient Department. There were 20 subjects with mixed normal and impaired vision, including cataracts, (pseudo)aphakia or posterior segment diseases. The six survey teams examined the same 20 individuals during the assessment. The other teams' findings were compared to those of the most senior or experienced team (as the gold standard).

For the IOV assessment, the Kappa coefficient values were classified as poor ($<0.20$), fair ($0.21$–$0.40$), moderate ($0.41$–$0.60$), good ($0.61$–$0.80$), and very good ($0.81$–$1.00$). RAAB methodology required agreement for each question to be $>0.60$.

The Kappa results for the Sarawak team were 37.0% for kappa value $\leq 0.60$ (28 out of 75 questions) and 63.0% for kappa value $>0.60$ (47 out of 75 questions) compared with the "gold standard" survey team. The Kappa results for the Eastern Region team were 18.7% for kappa value $\leq 0.60$ (14 out of 75 questions) and 81.3% for kappa value $>0.60$ (61 out of 75 questions) compared with the "gold standard" survey team. The questions and the corresponding team with kappa value $\leq 0.60$ were identified. Post-mortem in the form of interviews and group retraining focusing on the disagreements were done on the third day (after the IOV exercise) before the team went to the field the next day for an actual survey.

## Survey methods

Each region had one coordinator responsible for the smooth implementation and progress of the survey. Each team was assigned to survey 16–17 EBs. Subjects were selected from each block using the Compact Segment Sampling method. Following this method, the population area was divided into segments of equal population size, enough to provide the required number of eligible people aged 50 years and older. The survey started in a randomly selected segment until 50 eligible people had been examined. If all houses in the segment had been visited and the number of subjects was insufficient, then a second random segment was picked for the survey team to continue until they recruited 50 subjects.

If the subject was unavailable at home, the contact number would be taken from the neighbours, and a revisit would be done before the team left the survey area. If the subject could not be examined after three revisits, this person would be recorded as 'Not Available' and included in prevalence and cause analysis. Door-to-door interviews were conducted in each randomly selected EB. Subjects were recruited if they were 50 years and older, residing in the area for at least six months, and provided informed consent. Non-residents were excluded from the study.

Written consent from all participants was obtained, as stated in the ethics statement. Participants were provided with a consent form that clearly explained the study's purpose, procedures, potential risks, and their rights as participants. The consent form was written in a language that participants understood and they were given sufficient time to review and ask any questions before providing their written consent. In cases where verbal consent was obtained from illiterate respondents, we have documented and witnessed it according to the established protocol.

A total of 50 subjects were recruited in each EB. All recruited subjects had a brief interview, where demographic, medical and ocular history data were taken. It was followed by visual acuity assessment conducted at a distance of three meters using tablets installed with the RAAB7 application which incorporated built-in quality control measures through the PEEK Vision platform [8,9].

An eye examination was performed by the doctors in the survey team using a hand-held ophthalmoscope. The fundus examination was conducted using a direct ophthalmoscope after pupil dilatation with dilating eye drops, especially in cases where vision is poor (<6/12). The aim of the fundus examination was to look for abnormalities in the posterior segment of the eye which could potentially be the primary cause of visual impairment (main cause of presenting VA<6/12). The data collectors were trained to look for signs of myopic degeneration, glaucoma, diabetic retinopathy or age related macular degeneration.

Should subjects have a visual impairment, the primary cause was identified (following the RAAB protocol), and the subjects were referred to the nearest ophthalmic care facility for further management.

## Definition

The prevalence of distance VI was reported using Presenting Visual Acuity (PVA) in the better eye. It was derived from the Uncorrected Distant VA (UCVA) and Corrected Distance VA (CVA) inputs (if only UCVA available, PVA = UCVA; if UCVA and CVA available, PVA = CVA, assuming a habitually wearing). Pinhole VA (PinVA) would be tested if VA was worse than 6/12. VI categories were defined according to the Visual Acuity (VA) thresholds used in the WHO's International Classification of Diseases (ICD-11) [10]:

- Blindness: PVA less than 3/60 in the better eye

- Severe VI: PVA less than 6/60 to 3/60 in the better eye

- Moderate VI: PVA less than 6/18 to 6/60 in the better eye

- Mild VI: PVA less than 6/12 to 6/18 in the better eye

Once identified after eye examination, the primary cause of VI was categorized into treatable, preventable, avoidable, and posterior segment disease. Treatable VI were conditions that can be treated (for example, Refractive Error, Aphakia, and Cataract). Preventable VI were conditions that could be prevented by Primary Health Care (PHC), Primary Eye Care (PEC), or specialized ophthalmic services. Treatable and Preventable constituted Avoidable.

## Limitation

The doctors in the survey team (responsible for diagnosing ocular conditions and delivering medical explanation to the subjects when required) were the department's youngest medical doctors or ophthalmology trainees. They have varying ophthalmology experience and exposure to basic eye care training. Hence, there were disagreements during the inter-observation

assessment. Given the long duration of fieldwork, most hospitals in both regions could not allow more senior doctors to be recruited because of workforce shortage.

Data collection coincided with the pre-election time for the state legislative assemblies in both regions. It was impossible to reschedule the survey activities because the election date was only announced after all the logistics planning for the fieldwork, including the workforce mobilization, had been done. Although the highest level of permission to visit the houses, examine, and interview the subjects was applied and approved by the local authorities, there was resistance from certain subjects/communities to the examination/interview, alleging that the study had political intentions. The data collectors took more time to explain to the community leaders and subjects that this study had no political agenda. In several Enumeration Blocks (EBs), subjects had to be revisited before they agreed to participate. It resulted in a prolonged stay for the data collectors in the EB; hence, more money was spent on their accommodation and other logistics.

### Ethics approval

Medical Research and Ethics Committee of the Malaysian Ministry of Health (Research ID NMRR-19-197-46172). The study was conducted in accordance with the tenets of the Declaration of Helsinki.

### Sample size calculation

The latest population data was obtained from the Malaysian National Census 2020 [11,12]. A prevalence of blindness of 1.5% in Eastern and 1.6% in Sarawak among subjects aged 50 and older from NES II (2014) was used in the calculation using a 95% confidence interval, precision of 30% and estimated design effect (DEFF) of 1.5. It took into consideration the possibility of 20% non-response [5,6]. The calculation resulted in a sample size of 105 clusters (5239 subjects aged 50 years and older) and 98 clusters (4900 subjects aged 50 years and older) for Eastern and Sarawak, respectively.

### Statistical analysis

Data were entered into the cloud-based RAAB7 software using tablets. It reported the prevalence of VI in percentages and 95% Confidence Interval values by adjusting for age and sex. Other categorical data were reported in frequency and percentage. Raw data and digital reports were generated automatically in real-time and accessible to the authorized investigators through the web-based portal.

### Results

A total of 10,184 subjects 50 years old and older were enumerated: Eastern, n = 5,250, Sarawak, n = 4,934). A total of 9,709 were examined: Eastern, n = 4,961 (94.5% response), Sarawak, n = 4,748 (96.2% response). Of the 475 non-respondents, 73 (0.7%) refused to participate, 329 (3.3%) were incapable of being examined due to communication problems such as deafness or dementia, and 73 (0.8)% were not available. More female subjects were examined, n = 5,520 (56.8%) (Table 1).

The sample had less younger subjects in both genders (50–59 years old) than the survey area, but had more subjects in the older age group (except in the age group 80+ in Sarawak) (Fig 2). All the results, therefore, were analyzed using age-sex-adjusted estimates due to the difference.

**Table 1. Examination status, NES III (2023).**

| EASTERN | Female | | Male | | Total | |
|---|---|---|---|---|---|---|
| **Examination Status:** | n | % | n | % | n | % |
| Examined* | 2829 | 95.6 | 2132 | 93.0 | 4961 | 94.5 |
| Refused | 20 | 0.7 | 17 | 0.7 | 37 | 0.7 |
| Incapable | 83 | 2.8 | 105 | 4.6 | 188 | 3.6 |
| Unavailable | 26 | 0.9 | 38 | 1.7 | 64 | 1.2 |
| Total** | 2958 | 100.0 | 2292 | 100.0 | 5250 | 100.0 |
| SARAWAK | Female | | Male | | Total | |
| **Examination Status:** | n | % | n | % | n | % |
| Examined* | 2691 | 96.4 | 2057 | 96.0 | 4748 | 96.2 |
| Refused | 28 | 1.0 | 8 | 0.4 | 36 | 0.7 |
| Incapable | 67 | 2.4 | 74 | 3.5 | 141 | 2.9 |
| Unavailable | 6 | 0.2 | 3 | 0.1 | 9 | 0.2 |
| Total** | 2792 | 100.0 | 2142 | 100.0 | 4934 | 100.0 |

* Total examined (Percentage is the response rate).

** Total enumerated.

There was a higher prevalence of blindness in the Eastern than Sarawak: 0.8%, 95%CI (0.5, 1.1) vs 0.6%, 95%CI (0.3,0.9). However, the prevalence of blindness and prevalence of severe VI appeared lower, comparing the results to NES II (2014): Blindness Eastern from 1.4%, 95% CI (0.9, 1.9) to 0.8%, 95%CI (0.5, 1.1) and Blindness Sarawak from 1.6% 95%CI (1.0, 2.1) to 0.6%, 95%CI (0.3, 0.9). Severe VI Eastern from 1.2%, 95%CI (0.8, 1.7) to 0.9%, 95%CI (0.6, 1.1) and Severe VI Sarawak from 1.1% 95%CI (0.6, 1.6) to 0.9%, 95%CI (0.6, 1.2) (Table 2). There was no significant difference between males and females in any category of Visual Impairment during both surveys (Table 2).

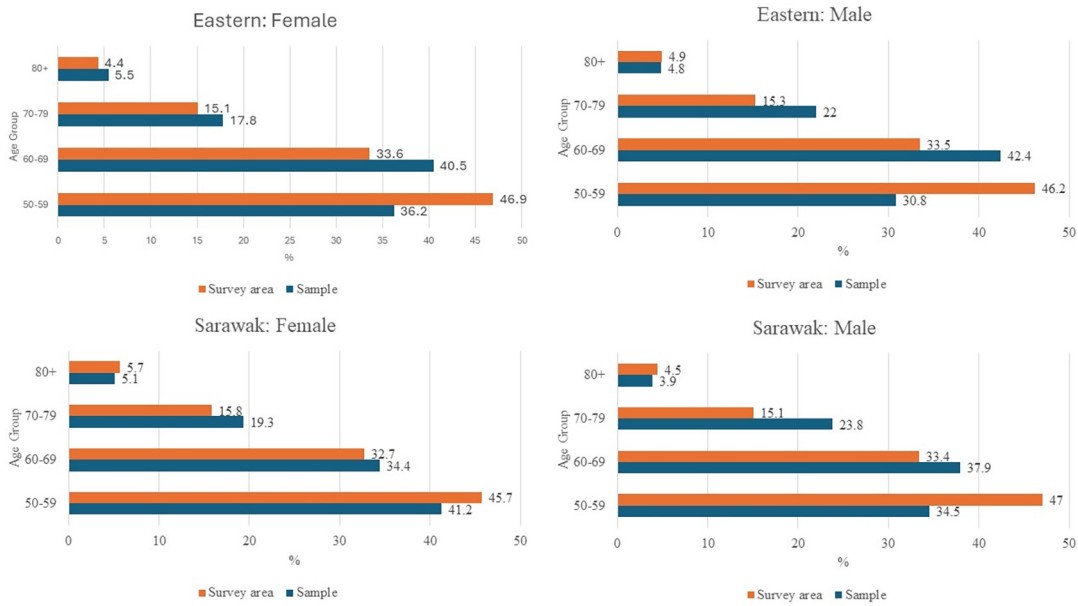

**Fig 2. Population in sample vs survey areas, NES III (2023).**

**Table 2. Adjusted prevalence and magnitude of blindness, severe, moderate, and mild vision impairment- comparing NES II (2014) and NES III (2023).**

| | NES II (2014) | | | | | | NES III (2023) | | | | | |
|---|---|---|---|---|---|---|---|---|---|---|---|---|
| **EASTERN** | Female | | Male | | Total | | Female | | Male | | Total | |
| **VI level:** | % | 95% CI | % | 95% CI | % | 95% CI | % | 95% CI | % | 95% CI | % | 95% CI |
| Blind | 1.6 | 1.0, 2.3 | 1.2 | 0.5, 1.9 | 1.4 | 0.9, 1.9 | 0.9 | 0.5, 1.2 | 0.7 | 0.3, 1.1 | 0.8 | 0.5, 1.1 |
| Severe | 1.4 | 0.7, 2.0 | 1.1 | 0.4, 1.8 | 1.2 | 0.8, 1.7 | 0.7 | 0.4, 1.1 | 0.9 | 0.5, 1.4 | 0.9 | 0.6, 1.1 |
| Moderate | 5.7 | 4.2, 7.2 | 4.7 | 3.2, 6.2 | 5.2 | 4.2, 6.3 | 7.2 | 6.0, 8.3 | 6.5 | 5.3, 7.6 | 6.7 | 5.8, 7.6 |
| Mild | 11.3 | 9.2, 13.5 | 8.2 | 6.2, 10.2 | 9.8 | 8.1, 11.6 | 7.1 | 6.1, 8.2 | 6.0 | 4.8, 7.3 | 6.4 | 5.5, 7.3 |
| **SARAWAK** | | | | | | | | | | | | |
| **VI level:** | % | 95% CI | % | 95% CI | % | 95% CI | % | 95% CI | % | 95% CI | % | 95% CI |
| Blind | 2.1 | 1.3, 3.0 | 1.0 | 0.4, 1.7 | 1.6 | 1.0, 2.1 | 0.8 | 0.4, 1.2 | 0.4 | 0.1, 0.8 | 0.6 | 0.3, 0.9 |
| Severe | 1.3 | 0.6, 2.0 | 0.9 | 0.3, 1.5 | 1.1 | 0.6, 1.6 | 0.9 | 0.5, 1.3 | 1.0 | 0.5, 1.4 | 0.9 | 0.6, 1.2 |
| Moderate | 7.8 | 5.8, 9.9 | 7.4 | 5.4, 9.4 | 7.6 | 5.8, 9.4 | 8.8 | 7.5, 10.1 | 6.8 | 5.3, 8.3 | 7.8 | 6.6, 9.0 |
| Mild | 11.5 | 8.9, 14.1 | 10 | 8.0, 11.9 | 10.7 | 8.8, 12.6 | 10.1 | 8.7, 11.5 | 8.8 | 7.3, 10.2 | 9.4 | 8.3, 10.6 |

CI = Confidence Interval; NES = National Eye Survey; VI = Visual Impairment.

The main cause of blindness was untreated cataracts: 77.3% (Eastern) and 75.0% (Sarawak). Although diabetic retinopathy was the 2nd leading cause of blindness in Eastern (9.1%) and severe VI (6.2%), it was not the cause of blindness but the 4th leading cause of severe VI in Sarawak (2.1%) (Table 3).

## Discussion

The availability of RAAB7 software facilitated data entry and cleaning during National Eye Survey III (NES III). The web platform with live survey data visibility enabled real-time monitoring for quality assurance throughout the survey. The trainer and investigators could identify and respond to any issues and questions from the data collectors on the ground in real time. Compared to NES II (2014), digitalization of data entry, cleaning, analysis and delivery of reports during NES III (2023) assured data quality. It saved time and, therefore, costs needed to support the survey teams on the ground as the duration of data collection was shortened.

Females formed 56.4% out of the 10,184 subjects enumerated. This population gender ratio was not consistent with the national census [11,12]. The greater representation of females in the survey could be influenced by traditional gender roles in Malaysia, where females are available because they are likely to stay at home taking care of the family's daily needs throughout the day while the males go out to work. The sample also had less younger subjects in both genders (50–59 years old) than the survey area, but had more subjects in the older age group (except in the age group 80+ in Sarawak). The younger individuals could have competing priorities, such as work commitments outside the house when the survey was done. This pattern is consistent and can be seen in other surveys [13–17].

The prevalence of blindness during the six simultaneous RAAB survey (NES II 2014) in Malaysia ranged from 0.5–1.9% (with a calculated weightage of 1.2%) [5]. The prevalence was lower, 0.8% and 0.6% for two regions (Eastern and Sarawak, respectively) during NES III in 2023. In general, the prevalence of VI during NES II and III were lower compared to other countries in the South East Asia Region [18–20]. In a global comparison of the prevalence of blindness among countries using the same RAAB methodology, Malaysia reported a lower prevalence in 2023, with the Sarawak zone at 0.6% and the Eastern zone at 0.8%. In contrast, Indonesia's Jawa Barat had a prevalence of 2.8% in 2014, Indonesia's Jawa Timur at 4.4%, Jawa Tengah at 2.7%, and Viet Nam Dinh at 2.1% in 2015 [21]. Malaysia's blindness prevalence was

**Table 3. Percentage of the principal causes of blindness and visual impairment in the Eastern Region and Sarawak, comparing NESII and NES III.**

| | NESII (2014) | | | | NESIII (2023) | | | |
|---|---|---|---|---|---|---|---|---|
| Principal cause | blind | severe VI | moderate VI | mild VI | blind | severe VI | moderate VI | mild VI |
| **EASTERN** | | | | | | | | |
| Uncorrected refractive error | 0.0 | 6.2 | 8.8 | 43.0 | 0.0 | 4.2 | 46.4 | 79.3 |
| Uncorrected aphakia | 0.0 | 0.0 | 0.7 | 0.0 | 0.0 | 0.0 | 0.0 | 0.0 |
| Untreated cataract | 60.0 | 68.8 | 79.6 | 46.8 | 77.3 | 81.2 | 45.7 | 17.4 |
| Cataract surgical complications | 2.5 | 0.0 | 2.0 | 2.3 | 0.0 | 0.0 | 1.0 | 0.0 |
| Pterygium | 0.0 | 6.2 | 0.7 | 0.4 | 0.0 | 0.0 | 0.0 | 0.0 |
| Other corneal opacity | 0.0 | 0.0 | 0.0 | 0.0 | 0.0 | 2.1 | 0.3 | 0.0 |
| Phthisis | 2.5 | 0.0 | 0.0 | 0.0 | 2.3 | 0.0 | 0.0 | 0.0 |
| Myopic Degeneration | 2.5 | 0.0 | 0.0 | 0.4 | 0.0 | 0.0 | 0.0 | 0.0 |
| Glaucoma | 12.5 | 0.0 | 0.7 | 0.4 | 4.5 | 2.1 | 1.8 | 0.0 |
| Diabetic retinopathy | 10.0 | 15.6 | 4.8 | 3.4 | 9.1 | 6.2 | 2.0 | 1.7 |
| Age-related macular degeneration | 0.0 | 0.0 | 0.7 | 2.6 | 0.0 | 0.0 | 0.8 | 0.3 |
| Other posterior segment disease | 7.5 | 3.1 | 0.0 | 0.4 | 4.5 | 4.2 | 0.8 | 0.6 |
| Other globe or CNS abnomalities | 2.5 | 0.0 | 2.0 | 0.4 | 0.0 | 0.0 | 0.0 | 0.0 |
| Total | 100.0 | 100 | 100 | 100 | 97.7 | 100 | 98.7 | 99.2 |
| Principal cause | blind | severe VI | moderate VI | mild VI | blind | severe VI | moderate VI | mild VI |
| **SARAWAK** | | | | | | | | |
| Uncorrected refractive error | 0.0 | 0.0 | 4.5 | 35.8 | 0.0 | 4.3 | 35.7 | 64.0 |
| Untreated cataract | 65.9 | 82.8 | 90.9 | 59.5 | 75.0 | 80.9 | 59.3 | 29.8 |
| Cataract surgical complications | 0.0 | 3.4 | 1.5 | 1.5 | 0.0 | 2.1 | 0.5 | 0.6 |
| Pterygium | 0.0 | 0.0 | 0.5 | 0.7 | 0.0 | 0.0 | 0.0 | 0.0 |
| Other corneal opacity | 2.4 | 6.9 | 0.0 | 0.0 | 3.1 | 0.0 | 0.0 | 0.0 |
| Phthisis | 4.9 | 0.0 | 0.0 | 0.0 | 6.2 | 0.0 | 0.0 | 0.0 |
| Myopic Degeneration | 0.0 | 0.0 | 0.0 | 0.4 | 0.0 | 0.0 | 0.0 | 0.0 |
| Glaucoma | 7.3 | 3.4 | 0.5 | 0.4 | 3.1 | 4.3 | 0.2 | 0.6 |
| Diabetic retinopathy | 2.4 | 3.4 | 1.0 | 0.0 | 0.0 | 2.1 | 1.7 | 2.1 |
| Age-related macular degeneration | 0.0 | 0.0 | 0.0 | 0.4 | 0.0 | 0.0 | 0.2 | 0.0 |
| Other posterior segment disease | 9.8 | 0.0 | 1.0 | 1.5 | 9.4 | 2.1 | 0.5 | 0.6 |
| Other globe or CNS abnomalities | 7.3 | 0.0 | 0.0 | 0.0 | 0.0 | 0.0 | 0.0 | 0.2 |
| Total | 100.0 | 100.0 | 100.0 | 100.0 | 96.9 | 95.7 | 98.1 | 97.9 |

CNS = Central Nervous System.

relatively comparable to the South East Asia Region (SEAR) contexts, indicating similar eye health challenges in the region (Nepal 1.1%, Bhutan 1.0%, India 2.0% and Cambodia 2.5%) [22–25]. It is essential to acknowledge the country's position regarding the prevalence compared to neighbouring countries, but achieving a reduction following an intervention is more important or meaningful.

Untreated refractive error as the second most common cause of moderate visual impairment in both regions could be attributed to various factors. Limited access to corrective eyewear, lack of awareness about refractive errors, and challenges in obtaining regular eye examinations may collectively contribute to the impact of refractive error on visual impairment.

There was an added workflow to check visual acuity in NESIII (refer to definition in methodology). This workflow was intended to help calculate the effective Refractive Error Coverage (eREC) [26]. In NESII, either PVA (with available correction) or PinVA (with available

correction) were tested. The increasing percentage of untreated refractive error was probably due to the increasing detection of refractive errors due to under/overcorrections or improper glasses fitting. Whether it was due to a rising prevalence of myopia among people 50 years and older, further research with proper refraction in the field needs to be done.

The possible reason on why cataract remained the most common causes of blindness and vision impairment was due to the survey protocol. The specific cause of poor vision (worse than 6/12) would be captured for individual eyes. However, if the cause was different between the eyes, the principal (main) cause of poor vision would be selected. The principal cause of poor vision was the condition more easily treated (to move the person from having VI to having normal vision). For example, in between diabetic retinopathy vs cataract, the principal cause of poor vision was cataract. The other possibility was because of the high percentage of untreated cataract in the blind category, the absence of fundus view did not allow for the detection of other avoidable blindness in the eye/person, hence the low prevalence of other avoidable blindness or VI.

In case of cataract, when comparing NES II to NES III, although the prevalence of blindness was lower in both regions during NES III, there was an increase in the percentage of untreated cataract as the cause of blindness in both regions and severe VI in the Eastern Region. The demographic population could partially explain these changes, i.e., the increased percentage of each age group 50 years and older (except for the 50–59 age group in the Eastern Region). The other possible explanation was that the amount of awareness and advocacy from the eye care delivery was still inadequate among the cataract population in the blind category. The percentages of severe, moderate and mild VI appeared to reduce in Sarawak. The percentages of moderate and mild VI also appeared to decrease in the Eastern Region.

In general, diabetic retinopathy as the cause for blindness and VI appeared to decrease except for moderate and mild VI in Sarawak. This improvement could be attributed to the national priorities and policies on Diabetes Mellitus, not KK-KKM initiatives [27]. The increase in the percentage of DR as the cause for moderate and mild VI in Sarawak could be associated with geographical access or coverage in DR screening within the state. Further research and analysis are needed as the issue is beyond the scope of this study.

The prevalence of DR resulting in visual impairment in Eastern was higher than in Sarawak. This is consistent with the findings of a higher prevalence of diabetes mellitus in Eastern than Sarawak [28]. We acknowledge that more work needs to be done at the Ministerial level with the Non-Communicable Disease division to strategically collaborate to curtail the increasing number of people with diabetes mellitus and prevent diabetic retinopathy through more active nationwide screening initiatives.

One of the national initiatives which can potentially be modified to incorporate for diabetic retinopathy screening is the *Klinik Katarak-Kementerian Kesihatan Malaysia* (Cataract Clinic Ministry of Health Malaysia, *KK-KKM*), an outreach arm of the ministry to reach the population (Fig 3). It was launched in 2014 in Sarawak and the Eastern Region of Malaysia as part of the country's progress commitments with the World Health Assembly (WHA) 66.4 resolution. It was also implemented as part of the national action plan following the findings of the NES II (2014). The KK-KKM uses buses to transport surgical and medical equipment along the selected service routes according to the location of Provincial Hospitals, which are used as the primary service sites. Once arrived at the site, the equipment is offloaded and used in the clinic (for screening) or operating room for (cataract surgery).

The service concept revolved around providing eye care services near patients' homes and optimizing existing facilities in provincial hospitals for cataract surgeries instead of operating in mobile units [29–31]. The KK-KKM project in both regions emphasized scheduled trips for screening, surgeries, and follow-up visits after one month to evaluate visual outcomes. A fixed

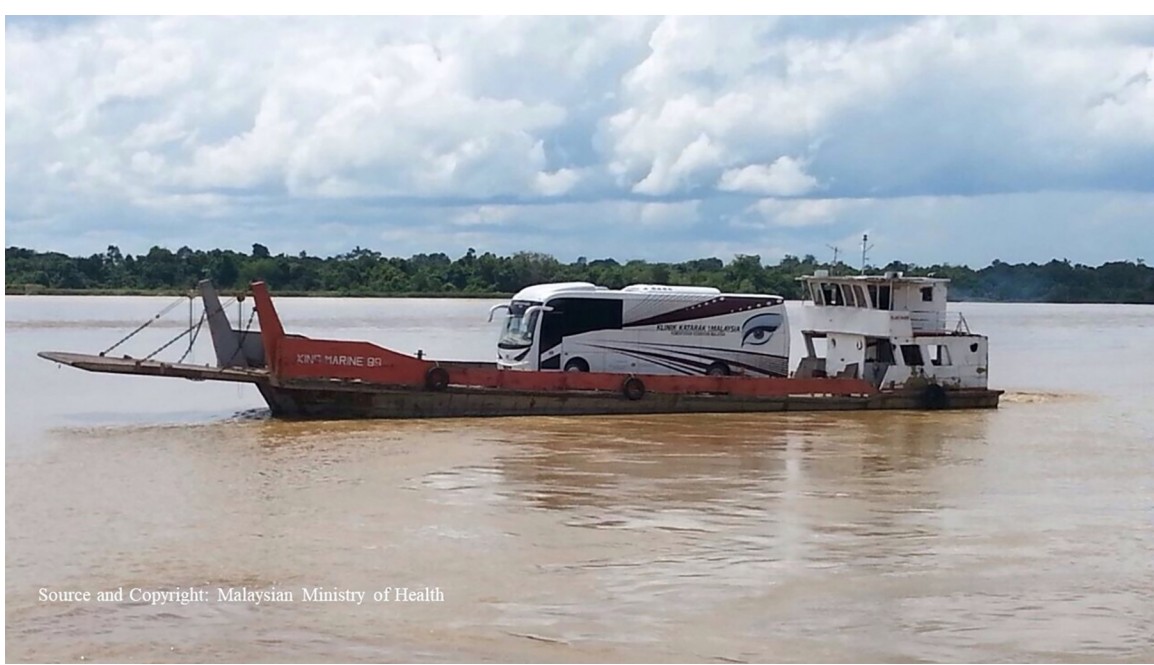

**Fig 3. The mobile unit on a boat en route to Kapit, a remote district in Sarawak.**

timetable was distributed to all provincial hospitals at the start of the year, enabling individuals in remote areas to plan their visits for eye treatment effectively. Although branded as a cataract surgical program, the screening platform and community engagement activities within the program encourage eye care advocacy within the community. KK-KKM allows screening of any individuals with any eye problems. They will be treated or referred to the nearest hospitals accordingly. It features a collaboration of multiple stakeholders and eye care providers from the community, Non-Governmental Bodies (NGOs), local Councils, Community Nurses, Primary Eye Care Doctors, Optometrists and Ophthalmologists. Data from cataract surgeries performed at the KK-KKM locations are also monitored centrally for the centre and individual surgeons's performance monitoring [32–39].

The concept of "Bringing High Impact Quality Eye Care Closer to Home", community engagement/advocacy and performance monitoring in outreach cataract surgery could have explained the reduction in the prevalence of blindness within both regions after nine years of service. The objective, concept and work process were endorsed by WHO when it was selected as a Case Study for the Western Pacific WHO Innovation Challenge in 2021/2022 [40]. Although implemented in Sarawak and the Eastern Region only, KK-KKM work concept was emulated by other regions. Many states started expanding their cataract service as outreach initiatives, not using dedicated mobile units but different methods, such as transporting patients to the main hospital or equipment to provincial hospitals using other smaller vehicles. The financial allocation, priorities, policies, and strategic plan for all cataract surgical services are under a single common governance (MOH). We postulate that positive results would also be seen in the other four regions if we ran the surveys concurrently.

## Conclusion

The prevalence of blindness and vision impairment was lower in 2023 compared to 2014 in both Sarawak and the Eastern Region of Malaysia. The prevalence and the changing trend of

the principal cause of blindness and VI could have been attributed to the KK-KKM, a cataract surgical outreach project operational at the community level which emphasized surgical outcomes as its service output. The project could further be utilized as a platform for multisectoral collaboration for community advocacy in cataract surgery and to address the high percentage of avoidable blindness within both regions.

## Acknowledgments

The authors would like to thank the Director General of the Ministry of Health Malaysia for his permission to publish this article. The authors would also like to acknowledge the contribution of data entry by the data collectors during both NES II and III.

## Author Contributions

**Conceptualization:** Mohamad Aziz Salowi, Nyi Nyi Naing, Nor Fariza Ngah.

**Data curation:** Mohamad Aziz Salowi, Nor Fariza Ngah.

**Formal analysis:** Mohamad Aziz Salowi, Wan Radziah Wan Nawang, Nor Fariza Ngah.

**Funding acquisition:** Mohamad Aziz Salowi.

**Investigation:** Mohamad Aziz Salowi, Nyi Nyi Naing, Norasyikin Mustafa, Wan Radziah Wan Nawang, Siti Nurhuda Sharudin.

**Methodology:** Mohamad Aziz Salowi, Nyi Nyi Naing, Norasyikin Mustafa, Wan Radziah Wan Nawang, Siti Nurhuda Sharudin.

**Project administration:** Mohamad Aziz Salowi, Nyi Nyi Naing, Norasyikin Mustafa, Wan Radziah Wan Nawang, Siti Nurhuda Sharudin, Nor Fariza Ngah.

**Resources:** Mohamad Aziz Salowi, Wan Radziah Wan Nawang, Siti Nurhuda Sharudin.

**Software:** Mohamad Aziz Salowi, Norasyikin Mustafa, Wan Radziah Wan Nawang, Siti Nurhuda Sharudin.

**Supervision:** Mohamad Aziz Salowi, Nyi Nyi Naing, Norasyikin Mustafa, Wan Radziah Wan Nawang, Siti Nurhuda Sharudin, Nor Fariza Ngah.

**Validation:** Mohamad Aziz Salowi, Nyi Nyi Naing, Norasyikin Mustafa, Wan Radziah Wan Nawang, Siti Nurhuda Sharudin, Nor Fariza Ngah.

**Visualization:** Mohamad Aziz Salowi, Nyi Nyi Naing, Norasyikin Mustafa, Wan Radziah Wan Nawang, Siti Nurhuda Sharudin, Nor Fariza Ngah.

**Writing – original draft:** Mohamad Aziz Salowi.

**Writing – review & editing:** Mohamad Aziz Salowi, Nyi Nyi Naing, Norasyikin Mustafa, Wan Radziah Wan Nawang, Siti Nurhuda Sharudin, Nor Fariza Ngah.

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
