## [Decision Letter · Decision Letter 0]

29 May 2024

PONE-D-24-04016Prevalence of visual impairment and its causes in adults aged 50 years and older: Estimates from the National Eye Surveys in MalaysiaPLOS ONE

Dear Dr. Naing,

Thank you for submitting your manuscript to PLOS ONE. After careful consideration, we feel that it has merit but does not fully meet PLOS ONE’s publication criteria as it currently stands. Therefore, we invite you to submit a revised version of the manuscript that addresses the points raised during the review process.

We look forward to receiving your revised manuscript.

Kind regards,

Srinivas Marmamula

Academic Editor

PLOS ONE

Journal Requirements:

Additional Editor Comments:

The reviewers have provided comments to improve the manuscript. Kindly review the comments and submit the revised version for consideration.

Reviewers' comments:

Reviewer's Responses to Questions

**Comments to the Author**

1. Is the manuscript technically sound, and do the data support the conclusions?

Reviewer #1: Yes

Reviewer #2: Yes

2. Has the statistical analysis been performed appropriately and rigorously? 

Reviewer #1: Yes

Reviewer #2: Yes

3. Have the authors made all data underlying the findings in their manuscript fully available?

Reviewer #1: Yes

Reviewer #2: Yes

4. Is the manuscript presented in an intelligible fashion and written in standard English?

Reviewer #1: Yes

Reviewer #2: Yes

5. Review Comments to the Author

Reviewer #1: This is an important paper and the data will provide useful feedback to eye service organizers in Malaysia. There are three items that need to be addressed as listed here:

1. Ethical approval information is listed in the initial application, but is not included in the report narrative itself. Please include the ethical approval information in the report narrative in the Methods section per journal requirements and as requested on the submission form (page. 5 of PDF package).

2. Inter-observer agreement information is found in three separate parts of the manuscript making it easy for the reader to gloss over this information and miss something important. You mention inter-observer agreement testing in the methodology (lines 106-110) and later list kappa values as being poor, fair, moderate, good or very good in a Definitions section (lines 152-153) and under the results section (lines 184-187) you provide a breakdown of inter-observer agreement scores by percent for fair, moderate, good and very good.

Please note that the typical convention is to discuss inter-observer agreement in the methodology section, including the Kappa statistic results. This is part of the preparation, thus methodology. Please also note that the RAAB manual lists Kappa statistic of 0.60 or greater as being acceptable. I do not understand what you mean when you provide a percentage breakdown. The way this section is written (lines 184-187) is appears that some of the results are below what is considered acceptable. If results are below what is acceptable (Kappa 0.60 or greater) were teams further trained and re-tested? If not, this would be a serious limitation that would also need to be discussed in a limitations section.

Please consolidate inter-observer agreement information in the methodology section. Please state clearly if the results were acceptable per RAAB standards. If results were not acceptable, please explain. If results were not acceptable also discuss the reasons and implications for this in the Limitations section.

3. Limitations - there is no section in the discussion for limitations (or strengths and limitations) of this study. This is a must. Your mention of the convenience of using the RAAB software package and digital data entry, etc. (lines 246-252) can be shortened and included as a strength. Please read several past RAAB publications for examples of what to include in the strengths and limitations section. Problems with inter-observer agreement not being up to standard should be discussed here as a limitation.

smaller items include:

Line 264-265 - you mention the prevalence of Vision impairment in your study is lower than other countries in the region and provide citations, but you don't actually provide any numbers for comparison. It would be useful to note what the range of Vision impairment is among the studies you are citing.

Conclusion - The first sentence "The prevalence of blindness and severe VI were reduced" is vague. I would make this sentence much more informative. For example "The prevalence of blindness and Vision impairment was lower in 2023 compared to 2014 in both the Sarawak region and the Eastern region."

Line 109 - this would be more clear if you stated "..and a test cluster in one of the nearby EB's during fieldwork."

Lines 137-139 you state. "Should subjects have a visual impairment, the primary cause was identified.........." Was the diagnosis of primary cause of vision impairment made via RAAB / WHO protocols? if so you should state this. If not, you should note why.

Lines 141-143. I did not understand what you mean when you state "......six survey teams were assess in their agreements asking 15 questions to subjects comprising patients and staff......" What questions? I don't understand this sentence.

Lines 176-177 you mention double data entry. As I recall, the standard RAAB survey package and use of tablets do not use double data entry.

Reviewer #2: This is a good article, in the sense that it reports prevalence of blindness and visual impairment after an intervention which followed a baseline study done in 2014.

However, there a few points to consider:

1. The term visual impairment, though not controversial in itself, should be distinguished from "blindness", in that the categories is defined in terms of Visual impairment. Hence, when referring to all vision loss categories, it is better to say "blindness and / or visual impairment".

2. There are a few typographical, grammatical or semantic errors that can be corrected, as suggested below:

Line 33: visual acuity "test" in stead of "check"

Line 76: "aimed" in stead of "aims"

Line 103: "Selected" in stead of "Individual"

Line 107: "Western Pacific" can be omitted

Line 109: "actual" can be replaced with "pilot"

Line 123: It should be made clear that the data so recorded will not be included in the prevalence and causes analysis

Line 125: Ordinarily, this should have meant a resident of that cluster or region, as non-resident participants will introduce bias.

Line 136: If this refers to Peek VA, it should be stated and appropriately referenced.

Line 155: There should be explanation why "best correction" is not used as measurement.

Line 190: The sentence starting with "These subjects..." can be omitted.

Line 246: NES III should be written in full at first mention in discussion.

Line 262: Insert "surveys" after RAAB.

Line 265: This is unclear, but can be improved by relating it to the context suggested in point 3.

Line 274: "strengthened" can be replaced with "modified to incorporate"

Line 280: The sentence can be modified as follows: "The KK-KKM" uses buses to transport..."

3. There is a need to provide context to the situation in the two regions of Malaysia, in order to make the article more palatable for a more regional and global audience. I would recommend one or two paragraphs (after the first paragraph of the Introduction) describing the current situation of blindness and visual impairment, first regionally (western Pacific), then Malaysia as a country, then the two study regions. This paragraph should summarise the epidemiology and service delivery aspects at the time of the study implementation.

4. The author must confirm which of the WHO or IAPB "owns' the RAAB protocol, and reference it appropriately (Line 81).

5. The training should summarise the main techniques covered in the training.

6. Elaboration explaining the fundus examination after dilatation should be included.

7. The section (starting from line 140) describing the IOV test should be part of the Training section. 8. The NES definitions starting at line 148 should be incorporated into the paragraph which starts at line 62.

9. The section starting from line 152 should be incorporated into the paragraph about the IOV test (see 6 above).

10. The IOV test is an internal quality control measure. It may or may not be considered as a study finding.

12. The explanation for greater representation of females could surmise other reasons from literature.

13. The explanation for lesser representation of participants aged under 60 years could surmise other reasons from literature.

14. It should be noted that the NES III prevalence of moderate visual impairment in both regions are higher than those in NES II, with possible explanation.

15. There should be reference made to the cause-specific changes over the two survey periods.

16. It should be noted that refractive error is second most prevalent cause of moderate visual impairment in both regions, with possible explanations.

17. The sections starting at line 287 and 295 are too expansive, should be omitted, but can be referred to as a link or appendix in the preceding paragraph.

18. Explanation should be proffered for the low prevalence of other avoidable causes of blindness and visual impairment, and why cataract is still the most common cause.

19. A postulate should be presented to suggest what impact could be anticipated from the surveys of the other 4 regions.

20. The conclusion should emphasize the gains made as evidence of this study, but also propose a way forward for eye care programming in Malaysia, based on the Discussion.

6. PLOS authors have the option to publish the peer review history of their article (what does this mean?). If published, this will include your full peer review and any attached files.

Reviewer #1: **Yes: **Jerry E Vincent

Reviewer #2: **Yes: **Dr Deon Minnies

---

## [Author Response · Author response to Decision Letter 0]

25 Jun 2024

PLOS ONE Journal 

Manuscript Title: Prevalence of visual impairment and its causes in adults aged 50 years and older: Estimates from the National Eye Surveys in Malaysia

Dear Srinivas Marmamula, 

Academic Editor

PLOS ONE

We would like to extend our sincere appreciation to the editor and reviewers for their insightful comments and valuable suggestions on our manuscript. We have modified the manuscript accordingly, and the detailed corrections are listed below point by point:

Reviewer #1

This is an important paper and the data will provide useful feedback to eye service organizers in Malaysia. There are three items that need to be addressed as listed here:

Comment 1:

Ethical approval information is listed in the initial application, but is not included in the report narrative itself. Please include the ethical approval information in the report narrative in the Methods section per journal requirements and as requested on the submission form (page. 5 of PDF package).

The ethical approval information has been added to the ‘Methods’ section, highlighted in red fonts, and incorporated into the manuscript. 

Line 215-218 : 

Ethics approval

Medical Research and Ethics Committee of the Malaysian Ministry of Health (Research ID NMRR-19-197-46172). The study was conducted in accordance with the tenets of the Declaration of Helsinki. 

Comment 2:

Inter-observer agreement information is found in three separate parts of the manuscript making it easy for the reader to gloss over this information and miss something important. You mention inter-observer agreement testing in the methodology (lines 106-110) and later list kappa values as being poor, fair, moderate, good or very good in a Definitions section (lines 152-153) and under the results section (lines 184-187) you provide a breakdown of inter-observer agreement scores by percent for fair, moderate, good and very good.

Please note that the typical convention is to discuss inter-observer agreement in the methodology section, including the Kappa statistic results. This is part of the preparation, thus methodology. Please also note that the RAAB manual lists Kappa statistic of 0.60 or greater as being acceptable. I do not understand what you mean when you provide a percentage breakdown. The way this section is written (lines 184-187) is appears that some of the results are below what is considered acceptable. If results are below what is acceptable (Kappa 0.60 or greater) were teams further trained and re-tested? If not, this would be a serious limitation that would also need to be discussed in a limitations section.

Please consolidate inter-observer agreement information in the methodology section. 

Please state clearly if the results were acceptable per RAAB standards. If results were not acceptable, please explain. If results were not acceptable also discuss the reasons and implications for this in the Limitations section.

We acknowledge the importance of consolidating this information in the methodology section, as per the typical convention. 

We have revised the manuscript. The assessment of inter-observer variation (IOV), the Kappa coefficient values and the results of IOV have been included in the ‘Survey Method’ section (under subheading ‘training’), highlighted in red fonts. (Line 127-137)

We have recalculated the values and presented them as ≤0.60 or >0.60. Kappa Tables for both Sarawak and the Eastern Region are also attached for your further perusal and decision whether they should be included in the main manuscript. 

Comment 3:

Limitations - there is no section in the discussion for limitations (or strengths and limitations) of this study. This is a must. Your mention of the convenience of using the RAAB software package and digital data entry, etc. (lines 246-252) can be shortened and included as a strength. Please read several past RAAB publications for examples of what to include in the strengths and limitations section. Problems with inter-observer agreement not being up to standard should be discussed here as a limitation.

The limitation section has been added in the main text, highlighted in red fonts. 

Line (197-214):

Limitation

The doctors in the survey team (responsible for diagnosing ocular conditions and delivering medical explanation to the subjects when required) were the department's youngest medical doctors or ophthalmology trainees. They have varying ophthalmology experience and exposure to basic eye care training. Hence, there were disagreements during the inter-observation assessment. Given the long duration of fieldwork, most hospitals in both regions could not allow more senior doctors to be recruited because of workforce shortage. 

Data collection coincided with the pre-election time for the state legislative assemblies in both regions. It was impossible to reschedule the survey activities because the election date was only announced after all the logistics planning for the fieldwork, including the workforce mobilization, had been done. Although the highest level of permission to visit the houses, examine, and interview the subjects was applied and approved by the local authorities, there was resistance from certain subjects/communities to the examination/interview, alleging that the study had political intentions. The data collectors took more time to explain to the community leaders and subjects that this study had no political agenda. In several Enumeration Blocks (EBs), subjects had to be revisited before they agreed to participate. It resulted in a prolonged stay for the data collectors in the EB; hence, more money was spent on their accommodation and other logistics. 

Other comments

Smaller items include:

Comment 1:

Line 264-265 - you mention the prevalence of Vision impairment in your study is lower than other countries in the region and provide citations, but you don't actually provide any numbers for comparison. It would be useful to note what the range of Vision impairment is among the studies you are citing.

Line 303-307: 

In a global comparison of the prevalence of blindness among countries using the same RAAB methodology, Malaysia reported a lower prevalence in 2023, with the Sarawak zone at 0.6% and the Eastern zone at 0.8%. In contrast, Indonesia's Jawa Barat had a prevalence of 2.8% in 2014, Indonesia's Jawa Timur at 4.4%, Jawa Tengah at 2.7%, and Viet Nam Dinh at 2.1% in 2015 [21].

Ref [21]: Rapid Assessment of Avoidable Blindness: Survey Data. Available at: https://www.raab.world/survey-data

Comment 2:

Conclusion - The first sentence "The prevalence of blindness and severe VI were reduced" is vague. I would make this sentence much more informative. For example "The prevalence of blindness and Vision impairment was lower in 2023 compared to 2014 in both the Sarawak region and the Eastern region."

This comment has been addressed in the manuscript, highlighted in red fonts.

Line 396-397: The prevalence of blindness and vision impairment was lower in 2023 compared to 2014 in both Sarawak and the Eastern Region of Malaysia. 

Comment 3:

Line 109 - this would be more clear if you stated "..and a test cluster in one of the nearby EB's during fieldwork."

This comment has been addressed in the manuscript, highlighted in red fonts. (Line 115-116) 

Comment 4:

Lines 137-139 you state. "Should subjects have a visual impairment, the primary cause was identified.........." Was the diagnosis of primary cause of vision impairment made via RAAB / WHO protocols? if so you should state this. If not, you should note why.

Line 173-175: Should subjects have a visual impairment, the primary cause was identified (following the RAAB protocol), and the subjects were referred to the nearest ophthalmic care facility for further management. 

Comment 5:

Lines 141-143. I did not understand what you mean when you state "......six survey teams were assess in their agreements asking 15 questions to subjects comprising patients and staff......" What questions? I don't understand this sentence.

For the Inter-observer Variation (IOV), the agreement was evaluated based on 15 questions related to vision and eye health assessments. These questions encompass aspects such as the use of spectacles for distance and near vision, acuity measurements for distance vision (presenting, corrected, pinhole, uncorrected), lens status for both eyes, and causes of poor vision for the right and left eyes. 

Table 1 and Table 2 present the summary of Kappa Values comparing the survey team to the gold standard team during the IOV assessment for RAAB in Sarawak and the Eastern Zone. Both tables will not be included in the main manuscript but will be submitted as ‘other’ for reference.

Table 1 Summary of Kappa Value comparing the survey team to the gold standard team during the Inter-observer Variation assessment for RAAB in Sarawak

No Question Team 1 95%CI Team 2 95%CI Team 3 95%CI Team 4 95%CI Team 5 95%CI 

1 spectacles_used_distance 0.95 0.86-1.04 0.85 0.69-1.01 0.85 0.69-1.01 0.90 0.77-1.03 0.90 0.77-1.03

2 spectacles_used_near 0.63 0.40-0.86 1.00 1.00-1.00 0.86 0.70-1.02 0.86 0.70-1.02 0.86 0.70-1.02

3 right_distance_acuity_presenting 0.54 0.29-0.79 0.59 0.35-0.83 0.63 0.40-0.86 0.58 0.34-0.82 0.53 0.28-0.78

4 left_distance_acuity_presenting 0.54 0.25-0.83 0.63 0.38-0.88 0.59 0.33-0.85 0.43 0.14-0.72 0.64 0.40-0.88

5 right_distance_acuity_corrected 0.83 0.67-0.99 0.74 0.55-0.93 0.78 0.60-0.96 0.87 0.73-1.01 0.75 0.56-0.94

6 left_distance_acuity_corrected 0.87 0.73-1.01 0.74 0.55-0.93 0.83 0.67-0.99 0.82 0.65-0.99 0.91 0.79-1.03

7 right_distance_acuity_pinhole 0.72 0.42-1.02 0.65 0.33-0.97 0.60 0.27-0.93 0.74 0.46-1.02 0.72 0.42-1.02

8 left_distance_acuity_pinhole 0.46 0.06-0.86 0.63 0.28-0.98 0.49 0.11-0.87 0.54 0.16-0.92 0.56 0.20-0.92

9 right_distance_acuity_uncorrected 0.58 0.38-0.78 0.68 0.50-0.86 0.61 0.41-0.81 0.53 0.32-0.74 0.67 0.48-0.86

10 left_distance_acuity_uncorrected 0.58 0.37-0.79 0.73 0.55-0.91 0.56 0.35-0.77 0.47 0.24-0.70 0.48 0.26-0.70

11 lens_status_right 0.69 0.43-0.95 0.71 0.44-0.98 0.62 0.34-0.90 0.65 0.37-0.93 0.71 0.44-0.98

12 lens_status_left 0.50 0.19-0.81 0.87 0.69-1.05 0.72 0.49-0.95 0.60 0.31-0.89 0.42 0.08-0.76

13 poor_vision_cause_right 0.72 0.51-0.93 0.54 0.29-0.79 0.64 0.41-0.87 0.73 0.53-0.93 0.62 0.38-0.86

14 poor_vision_cause_left 0.50 0.21-0.79 0.69 0.46-0.92 0.59 0.34-0.84 0.54 0.27-0.81 0.60 0.35-0.85

15 poor_vision_cause_principle 0.65 0.36-0.94 0.70 0.42-0.98 0.65 0.36-0.94 0.59 0.29-0.89 0.57 0.25-0.89

Compared to the “gold standard” team: 

≤0.60 = 28/75 questions = 37%

>0.60 = 47/75 questions = 63%

Table 2 Summary of Kappa Value comparing the survey team to the gold standard team during the Inter-observer Variation assessment for RAAB in the Eastern Region 

No Question Team 1 95%CI Team 2 95%CI Team 3 95%CI Team 4 95%CI Team 5 95%CI

1 spectacles_used_distance 0.86 0.70-1.02 0.91 0.79-1.03 1.00 1.00-1.00 0.91 0.79-1.03 0.96 0.87-1.05

2 spectacles_used_near 0.73 0.53-0.93 0.60 0.35-0.85 0.69 0.48-0.90 0.65 0.43-0.87 0.82 0.65-0.99

3 right_distance_acuity_presenting 0.68 0.46-0.90 0.48 0.21-0.75 0.63 0.39-0.87 0.44 0.17-0.71 0.68 0.46-0.90

4 left_distance_acuity_presenting 0.62 0.38-0.86 0.75 0.54-0.96 0.80 0.61-0.99 0.81 0.63-0.99 0.79 0.59-0.99

5 right_distance_acuity_corrected 0.83 0.67-0.99 0.80 0.64-0.96 0.96 0.88-1.04 0.80 0.63-0.97 0.84 0.69-0.99

6 left_distance_acuity_corrected 0.87 0.73-1.01 0.84 0.69-0.99 0.96 0.88-1.04 0.92 0.81-1.03 0.88 0.75-1.01

7 right_distance_acuity_pinhole 0.93 0.79-1.07 0.78 0.54-1.02 0.85 0.65-1.05 0.51 0.17-0.85 0.70 0.42-0.98

8 left_distance_acuity_pinhole 0.58 0.23-0.93 0.73 0.43-1.03 0.82 0.58-1.06 0.90 0.71-1.09 0.73 0.43-1.03

9 right_distance_acuity_uncorrected 0.65 0.47-0.83 0.51 0.30-0.72 0.64 0.45-0.83 0.50 0.29-0.71 0.66 0.47-0.85

10 left_distance_acuity_uncorrected 0.74 0.58-0.90 0.64 0.45-0.83 0.77 0.61-0.93 0.71 0.54-0.88 0.74 0.57-0.91

11 lens_status_right 0.94 0.82-1.06 0.53 0.18-0.88 0.78 0.57-0.99 0.76 0.54-0.98 0.42 0.02-0.82

12 lens_status_left 0.87 0.70-1.04 0.47 0.07-0.87 0.68 0.44-0.92 0.82 0.62-1.02 0.58 0.23-0.93

13 poor_vision_cause_right 0.73 0.53-0.93 0.52 0.26-0.78 0.67 0.45-0.89 0.49 0.23-0.75 0.55 0.30-0.80

14 poor_vision_cause_left 0.61 0.37-0.85 0.70 0.48-0.92 0.69 0.46-0.92 0.76 0.56-0.96 0.63 0.38-0.88

15 poor_vision_cause_principle 0.65 0.39-0.91 0.77 0.55-0.99 0.76 0.53-0.99 0.75 0.52-0.98 0.64 0.37-0.91

Compared to the “gold standard” team: 

≤0.60 = 14/75 questions = 18%

 >0.60 = 61/75 questions = 81.3% 

Comment 6:

Lines 176-177 you mention double data entry. As I recall, the standard RAAB survey package and use of tablets do not use double data entry.

Inconsistency check and double data entry were the manual features of RAAB6. What we meant by the statement was, RAAB7 includes this capability in the software automatically, therefore enhancing the quality and integrity of the collected data. We apologize, we decided to omit this statement to avoid confusion.

Reviewer #2

This is a good article, in the sense that it reports prevalence of blindness and visual impairment after an intervention which followed a baseline study done in 2014. However, there a few points to consider:

Comment 1:

The term visual impairment, though not controversial in itself, should be distinguished from "blindness", in that the categories is defined in terms of Visual impairment. Hence, when referring to all vision loss categories, it is better to say "blindness and / or visual impairment".

This comment has been addressed in the main text, highlighted in red fonts.

Comment 2:

There are a few typographical, grammatical or semantic errors that can be corrected, as suggested below:

Comment:

Line 33: visual acuity "test" in stead of "check"

This comment has been addressed in the main text, highlighted in red fonts. 

Line 32-35:

The prevalence of blindness and/or visual impairment (blindness, severe, moderate, and early) and its primary cause were determined through a visual acuity test and eye examination with a hand-held ophthalmoscope.

Comment:

Line 76: "aimed" in stead of "aims"

This comment has been addressed in the main text, highlighted in red fonts. 

Line 83-84:

This study aimed to determine the difference or improvement in the prevalence of blindness and/or visual impairment between the two administrative regions from the previous survey.

Comment:

Line 103: "Selected" in stead of "Individual"

This comment has been addressed in the main text, highlighted in red fonts. 

Line 108-109:

Selected EB codes and the corresponding maps were then used to identify the location of the EBs during fieldwork data collection.

Comment:

Line 107: "Western Pacific" can be omitted

This comment has been addressed in the main text

Line 113-114:

Those training sessions by a certified RAAB trainer were to ensure data quality and strict adherence to study protocol. 

Comment:

Line 109: "actual" can be replaced with "pilot"

This comment has been addressed in the main text, highlighted in red fonts. 

Line 114-116:

Survey team members were required to attend four training days as preparation, including RAAB lectures, inter-observer variation assessment and a pilot survey in a test cluster in one of the nearby EB's during fieldwork.

Comment:

Line 123: It should be made clear that the data so recorded will not be included in the prevalence and causes analysis

This comment has been addressed in the main text, highlighted in red fonts.

Line 149-151: 

If the subject could not be examined after three revisits, this person would be recorded as `Not Available'

Comment:

Line 125: Ordinarily, this should have meant a resident of that cluster or region, as non-resident participants will introduce bias.

This comment has been addressed in the main text, highlighted in red fonts.

Line 151-153: 

Subjects were recruited if they were 50 years and older, residing in the area for at least six months, and provided informed consent. Non-residents were excluded from the study.

Comment:

Line 136: If this refers to Peek VA, it should be stated and appropriately referenced.

Line 162-165: 

It was followed by visual acuity assessment conducted at a distance of three meters using tablets installed wi

---

## [Decision Letter · Decision Letter 1]

27 Aug 2024

PONE-D-24-04016R1Prevalence of visual impairment and its causes in adults aged 50 years and older: Estimates from the National Eye Surveys in MalaysiaPLOS ONE

Dear Dr. Naing,

Thank you for submitting your manuscript to PLOS ONE. After careful consideration, we feel that it has merit but does not fully meet PLOS ONE’s publication criteria as it currently stands. Therefore, we invite you to submit a revised version of the manuscript that addresses the points raised during the review process.

The manuscript requires minor revision before it can be accepted. Authors are requested to go through the reviewers comments and modify the manuscript accordingly.

We look forward to receiving your revised manuscript.

Kind regards,

Kumar Saurabh

Academic Editor

PLOS ONE

Journal Requirements:

Additional Editor Comments:

The manuscript requires minor revision before it can be accepted. Authors are requested to go through the reviewers comments and modify the manuscript accordingly.

Reviewers' comments:

Reviewer's Responses to Questions

**Comments to the Author**

1. If the authors have adequately addressed your comments raised in a previous round of review and you feel that this manuscript is now acceptable for publication, you may indicate that here to bypass the “Comments to the Author” section, enter your conflict of interest statement in the “Confidential to Editor” section, and submit your "Accept" recommendation.

Reviewer #2: All comments have been addressed

Reviewer #3: All comments have been addressed

2. Is the manuscript technically sound, and do the data support the conclusions?

Reviewer #2: Yes

Reviewer #3: (No Response)

3. Has the statistical analysis been performed appropriately and rigorously? 

Reviewer #2: Yes

Reviewer #3: Yes

4. Have the authors made all data underlying the findings in their manuscript fully available?

Reviewer #2: Yes

Reviewer #3: Yes

5. Is the manuscript presented in an intelligible fashion and written in standard English?

Reviewer #2: Yes

Reviewer #3: Yes

6. Review Comments to the Author

Reviewer #2: The authoors adequately addressed the comments provided by this reviewer. Two minor considerations should be made:

1. Verification from the authors and editor that the manuscript does not exceed the word count requirement of the publication.

2. The explanatory sentence starting in line 186 can be removed. The comment arose from the notion that earlier versions of RAAB conventionally used pinhole as best corrected. https://www.raab.world/analysis/uncorrected-va-and-erec-raab.

This reviewer does not need to see a next revision, and would be happy to see the article published.

Reviewer #3: I thank the authors for appropriately addressing all queries. I would also request that other national level eye surveys in the WHO SEAR region such as Nepal, India, Bhutan and Cambodia may be included in discussion apart from the Indonesian data. That will provide useful context to the situation in entire WHO-SEAR region.

Some relevant primary refernces are:

Lepcha NT, Sharma IP, Sapkota YD, Das T, Phuntsho T, Tenzin N, et al. Changing trends of blindness, visual impairment and cataract surgery in Bhutan: 2009-2018. PLoS ONE 2019;14(5):e0216398.

Mishra SK, Shah R, Gogate P, Sapkota YD, Gurung R, Shrestha MK, et al. Prevalence of blindness and vision impairment among people 50 years and older in Nepal: a national Rapid Assessment of Avoidable Blindness survey [Internet]. 2024 [cited 2024 Aug 11];Available from: http://medrxiv.org/lookup/doi/10.1101/2024.08.06.24311588

Vashist P, Senjam SS, Gupta V, Gupta N, Shamanna BR, Wadhwani M, et al. Blindness and visual impairment and their causes in India: Results of a nationally representative survey. PLoS One 2022;17(7):e0271736.

Cambodia 2019 - https://www.raab.world/survey/cambodia-2019

7. PLOS authors have the option to publish the peer review history of their article (what does this mean?). If published, this will include your full peer review and any attached files.

Reviewer #2: **Yes: **Deon Minnies

Reviewer #3: **Yes: **Dr Vivek Gupta, MD, MSc (PHEC)

---

## [Author Response · Author response to Decision Letter 1]

7 Sep 2024

PLOS ONE Journal

Manuscript Tittle: Prevalence of visual impairment and its causes in adults aged 50 years and older: Estimates from the National Eye Surveys in Malaysia.

Dear 

Kumar Saurabh

Academic Editor

PLOS ONE 

Thank you for considering our manuscript for publication in PLOS ONE. We appreciate the time and effort that the reviewers and editorial team have put into evaluating our submission. We have modified the manuscript accordingly and the detailed corrections are listed below point by point:

Editor comments on journal requirements:

Response to editor:

We believe that our reference lists are complete and correct, and none of them have been officially retracted.

Reviewer #2: 

Comment 

The authors adequately addressed the comments provided by this reviewer. Two minor considerations should be made:

1. Verification from the authors and editor that the manuscript does not exceed the word count requirement of the publication.

Response to comment:

We have reviewed the manuscript and confirmed that it adheres to the journal's guidelines regarding word count, number of figures, and supporting information. As per the journal's instructions, there are no restrictions on these elements. We have ensured that the manuscript is presented in a concise and clear manner, while providing sufficient detail to support our findings.

2. The explanatory sentence starting in line 186 can be removed. The comment arose from the notion that earlier versions of RAAB conventionally used pinhole as best corrected.

 https://www.raab.world/analysis/uncorrected-va-and-erec-raab.

Response to comment: 

We agree with the reviewer’s observation and have removed the explanatory sentence starting in line 186 from the revised manuscript. 

Line 186 

The best correction would require refraction, which would be time-consuming and impossible in the field without proper equipment and expertise. It was also not measured because it would deviate from RAAB protocol, which only measured corrected VA (using a pinhole).

Reviewer #3: 

Comment

I thank the authors for appropriately addressing all queries. I would also request that other national level eye surveys in the WHO SEAR region such as Nepal, India, Bhutan and Cambodia may be included in discussion apart from the Indonesian data. That will provide useful context to the situation in entire WHO-SEAR region.

Some relevant primary references are:

1.Lepcha NT, Sharma IP, Sapkota YD, Das T, Phuntsho T, Tenzin N, et al. Changing trends of blindness, visual impairment and cataract surgery in Bhutan: 2009-2018. PLoS ONE 2019;14(5):e0216398.

2.Mishra SK, Shah R, Gogate P, Sapkota YD, Gurung R, Shrestha MK, et al. Prevalence of blindness and vision impairment among people 50 years and older in Nepal: a national Rapid Assessment of Avoidable Blindness survey [Internet]. 2024 [cited 2024 Aug 11]; Available from: http://medrxiv.org/lookup/doi/10.1101/2024.08.06.24311588

3.Vashist P, Senjam SS, Gupta V, Gupta N, Shamanna BR, Wadhwani M, et al. Blindness and visual impairment and their causes in India: Results of a nationally representative survey. PLoS One 2022;17(7):e0271736.

4.Cambodia 2019 - https://www.raab.world/survey/cambodia-2019

Response to comment:

Thank you for your valuable feedback and for highlighting the importance of a broader regional context. We have incorporated data from these countries into our discussion to provide a more holistic view of the eye health landscape within WHO-SEAR, highlighted in RED font. (Line 307-310)

Line 307-310

Malaysia’s blindness prevalence was relatively comparable to the South East Asia Region (SEAR) contexts, indicating similar eye health challenges in the region (Nepal 1.1%, Bhutan 1.0%, India 2.0% and Cambodia 2.5%) [22-25].

References

22.Lepcha NT, Sharma IP, Sapkota YD, Das T, Phuntsho T, Tenzin N, et al. Changing Trends of Blindness, Visual Impairment and Cataract Surgery in Bhutan: 2009-2018. PLoS One. 2019;14(5):e0216398. DOI: 10.1371/journal.pone.0216398 PMID: 31071127

23.Mishra SK, Shah R, Gogate P, Sapkota YD, Gurung R, Shrestha MK, et al. Prevalence of Blindness and Vision Impairment Among People 50 years and Older in Nepal: A National Rapid Assessment of Avoidable Blindness Survey [Internet]. 2024 [cited 2024 Aug 11];Available from: http://medrxiv.org/lookup/doi/10.1101/2024.08.06.24311588. DOI: https://doi.org/10.1101/2024.08.06.24311588

24.Vashist P, Senjam SS, Gupta V, Gupta N, Shamanna BR, Wadhwani M, et al. Blindness and Visual Impairment and Their Causes in India: Results of A Nationally Representative Survey. PLoS One. 2022;17(7):e0271736. DOI: 10.1371/journal.pone.0271736 PMID: 35862402

25.RAAB.(2019). Cambodia 2019 Survey Data. Retrieved from https: //www.raab.world/survey/cambodia-2019

---

## [Decision Letter · Decision Letter 2]

8 Oct 2024

Prevalence of visual impairment and its causes in adults aged 50 years and older: Estimates from the National Eye Surveys in Malaysia

PONE-D-24-04016R2

Dear Dr. Naing,

We’re pleased to inform you that your manuscript has been judged scientifically suitable for publication and will be formally accepted for publication once it meets all outstanding technical requirements.

Kind regards,

Kumar Saurabh

Academic Editor

PLOS ONE

Additional Editor Comments (optional):

Reviewers' comments:

Reviewer's Responses to Questions

**Comments to the Author**

1. If the authors have adequately addressed your comments raised in a previous round of review and you feel that this manuscript is now acceptable for publication, you may indicate that here to bypass the “Comments to the Author” section, enter your conflict of interest statement in the “Confidential to Editor” section, and submit your "Accept" recommendation.

Reviewer #2: All comments have been addressed

Reviewer #3: All comments have been addressed

2. Is the manuscript technically sound, and do the data support the conclusions?

Reviewer #2: Yes

Reviewer #3: Yes

3. Has the statistical analysis been performed appropriately and rigorously? 

Reviewer #2: Yes

Reviewer #3: Yes

4. Have the authors made all data underlying the findings in their manuscript fully available?

Reviewer #2: Yes

Reviewer #3: Yes

5. Is the manuscript presented in an intelligible fashion and written in standard English?

Reviewer #2: Yes

Reviewer #3: Yes

6. Review Comments to the Author

Reviewer #2: All comments have been addressed in a previous submission. This reviewer has no further comments to the author.

Reviewer #3: Thank You for the edits. All comments addressed by the reviewers have been addressed satisfactorily by the authors

7. PLOS authors have the option to publish the peer review history of their article (what does this mean?). If published, this will include your full peer review and any attached files.

Reviewer #2: **Yes: **Dr Deon Minnies, Community Eye Health Institute, University of Cape Town, South Africa (d.minnies@uct.ac.za)

Reviewer #3: **Yes: **Dr Vivek Gupta, MD, MSc(PHEC)

---

## [Editor Report · Acceptance letter]

14 Oct 2024

PONE-D-24-04016R2 

PLOS ONE

Dear Dr. Naing, 

I'm pleased to inform you that your manuscript has been deemed suitable for publication in PLOS ONE. Congratulations! Your manuscript is now being handed over to our production team.

Kind regards, 

on behalf of

Dr. Kumar Saurabh 

Academic Editor

PLOS ONE